# The Navigation System of a Logistics Inspection Robot Based on Multi-Sensor Fusion in a Complex Storage Environment

**DOI:** 10.3390/s22207794

**Published:** 2022-10-14

**Authors:** Yang Zhang, Yanjun Zhou, Hehua Li, Hao Hao, Weijiong Chen, Weiwei Zhan

**Affiliations:** 1College of Economic and Management, Shanghai Polytechnic University, Shanghai 201209, China; 2Logistics Research Center, Shanghai Maritime University, Shanghai 201306, China

**Keywords:** robot navigation, sensor fusion, multiline lidar, depth camera, logistics warehousing

## Abstract

To reliably realize the functions of autonomous navigation and cruise of logistics robots in a complex logistics storage environment, this paper proposes a new robot navigation system based on vision and multiline lidar information fusion, which can not only ensure rich information and accurate map edges, but also meet the real-time and accurate positioning and navigation in complex logistics storage scenarios. Simulation and practical verification showed that the robot navigation system is feasible and robust, and overcomes the problems of low precision, poor robustness, weak portability, and difficult expansion of the mobile robot system in a complex environment. It provides a new idea for inspection in an actual logistics storage scenario and has a good prospective application.

## 1. Introduction

In recent years, with the arrival of Industry 4.0, mobile robot technology has also developed rapidly. At present, it has been widely used in logistics, manufacturing, agriculture, service, and other fields [1]. Among them, navigation is the key piece of mobile robot technology, mainly including slam and path planning. Slam refers to the positioning and mapping of the robot, and path planning is to plan a feasible path for the robot to avoid dynamic obstacles in the process of moving according to the optimization criteria.

Robot navigation technology began in 1972. Stanford University developed the first mobile robot that can make autonomous decisions and plan through the observations made by cameras, lidar, and other sensors. Since then, many research groups and scholars have carried out a lot of research on various navigation problems. At present, there are two main methods based on vision and lidar. For example, the mono slam, the first monocular slam system proposed by Smith [2] and others, uses a Kalman filter as the backend to track the sparse feature points at the frontend. Another example is the orb designed by Mur et al. The SLAM algorithm [3] solves the problem of cumulative error. Grisetti [4] et al. improved the slam method based on the Rao Black well particle filter and realized the gmapping algorithm. Eitan et al. [5] proposed an effective voxel-based 3D mapping algorithm, which can explicitly model the unknown space. The Rtabmap system proposed by Labb et al. [6] uses RGB-d cameras for synchronous positioning and local mapping to overcome the shortcomings of loop detection affecting real-time processing over time.

Robot patrol inspection in a storage environment can greatly improve the efficiency of logistics operations. However, the existing methods and technologies still have some problems, such as the poor autonomous navigation performance of mobile robots in complex scenes. For this reason, we designed a mobile robot autonomous mapping and path planning system based on multi-sensor fusion, which obtains the information on the surrounding environment through the 3D lidar, and realizes the autonomous positioning and mapping of the robot by using the real-time appearance mapping algorithm. To improve the efficiency of global path planning, we used the improved A* algorithm as the global path planning method, and the local path planning method uses the classic DWA algorithm to avoid obstacles.

## 2. System Framework

The multi-sensor information fusion logistics robot navigation system designed in this paper for the real and complex logistics warehousing scene is shown in Figure 1. Firstly, the 3D laser point cloud was denoised, downsampled, point cloud segmented, ground fit, and point cloud converted, and then the visual and lidar information was fused to use the rtabmap algorithm [6] for map modeling. Finally, DWA [7] and the improved A* algorithm were selected for local and global planning, respectively.

## 3. LIDAR Point Cloud Preprocessing

### 3.1. Point Cloud Filtering

In the process of lidar point cloud data acquisition, due to factors such as low equipment accuracy and a complex environment, there will be noise in the collected data, as shown in Figure 2. Therefore, we chose direct, statistical [8], and conditional filtering [9] to process noise. In addition, the amount of directly acquired point cloud data is large. In order to speed up the subsequent mapping, positioning, and other operations, voxel filtering [10] is also used to realize down sampling.

Pass-through filtering is to cut the outliers in the specified coordinate range, which can realize fast cutting of outliers. Statistical filtering is used to eliminate the obvious outliers introduced by noise through the comparison of mean and variance. The conditional rate is set to be similar to the piecewise function for targeted filtering. Voxel filtering is to desample lidar point cloud data for subsequent processing. After filtering and downsampling, the lidar point cloud is shown in Figure 3.

### 3.2. Ground Treatment

There is ground information in the filtered point cloud. In this study, an incremental line fitting algorithm [11] was used to segment the ground. The specific operations were as follows: converting the 3D point cloud (x,y,z) to (x,y) under the 2D plane, and partition. The space was divided into *N* parts, as shown in Figure 4, according to lidar characteristics, as shown in Formula (1).
(1)$$N=\frac{2\pi }{△\alpha }$$
where ▵α is the angle covered by each segment, and then $p_{i}$, the corresponding space groups, are
(2)$$segment\left(p_{i}\right)=\frac{\arctan(\frac{y_{i}}{x_{i}})}{▵\alpha }$$

## 4. Mapping and Navigation

### 4.1. Drawing Construction

For more accurate real-time mapping, the navigation mapping system chooses to build a map based on the real-time mapping rtabmap algorithm [12]. It mainly includes a front-end visual odometer, pose optimization, closed-loop detection, and mapping. The details are as follows: Obtain a visual odometer by comparing adjacent frames, calculate how the camera moves between frames, and build a local map at the same time. Then, it accepts the camera pose calculated by the visual odometer at different times, and optimizes the data obtained by loop detection to obtain a globally consistent camera path and environment map.

### 4.2. Planning Algorithm

To solve the problem that the standard A* global path planning algorithm takes too long, the global planning algorithm in this paper is based on the A* algorithm [13], cancels the close set, and modifies the open set searched from the starting node and the open set searched from the target point. It also modifies the heuristic function for Manhattan distance, as shown in Formula (3).
(3)$$h\left(n\right)=|x_{en}-x_{sn}|+|y_{en}-y_{sn}|$$

In the formula, $\left(x_{en},y_{en}\right)$ represents the current node coordinate in the starting direction from the starting point, and $\left(x_{sn},y_{sn}\right)$ represents the current point coordinate in the starting direction from the target point. It increases the weight of the heuristic function when calculating the total valuation, as shown in Formulas (4) and (5).
(4)f(n)=g(n)+t∗h(n)
(5)$$t=1+\frac{1}{l_{map}+W_{map}}$$
where *T* represents the sparse weight of the heuristic function, $l_{map}$ indicates the length of the map, and $w_{map}$ represents the width of the map. To solve the problem that A* is not suitable for dynamic obstacles, we selected the DWA algorithm [13] for local planning from the perspective of the actual navigation scene requirements of logistics robots. The specific algorithm works as follows: firstly, simulate the motion model of the logistics robot [14], as shown in Formula (6):(6)$$r_{c}=\frac{v_{c}}{w_{c}}=\frac{\left(v_{l}+v_{r}\right)d_{LR}}{2(v_{r}-v_{l})}$$
where $d_{LR}$ represents virtual wheel spacing; $v_{l}$ and $v_{r}$ represent the linear velocities of the virtual left and right tracks, respectively. The velocity is sampled under the constraint condition of the formula, and then the motion trajectory is calculated:(7)$$v_{m}=\left(v,m\right)|v\in \left[v_{min},v_{max}\right],w_{m}\in \left[v_{min},v_{max}\right]$$
where $v_{m}$ represents the possible speed space of the mobile robot; $v_{min}$ and $v_{max}$ represent the minimum and maximum linear speed that the robot can reach; $w_{min}$ and $w_{max}$ indicate the minimum angular velocity and the maximum linear velocity.
(8)$$v_{d}=\left\{\begin{array}{c} \left(v,m\right)|v\in \left[v_{c}-a_{max}\Delta t,v_{c}+a_{max}\Delta t\right]\\ w\in [w_{c}-\alpha_{max}\Delta t,w_{c}+\alpha_{max}\Delta t] \end{array}\right.$$
where $v_{d}$ represents the actual speed that the mobile robot can reach; $a_{max}$ and $\alpha_{max}$ represent the maximum acceleration and the maximum angular acceleration of the current mobile robot; Δt represents the cycle time interval of the mobile robot’s movement.
(9)$$v_{a}=\left(v,m\right)|v\leq \sqrt{sdist\left(v,m\right)a_{b}},w\leq \sqrt{2dist\left(v,m\right)\alpha_{b}}$$
where $v_{a}$ represents the safe speed of the mobile robot; dist(v,m) represents the minimum distance from the obstacle at a certain speed with a the corresponding curve of the track; $a_{b}$ and $\alpha_{b}$ represent the angular velocity and angular acceleration that can stop on this curved track. The final velocity space is formed by the intersection of three velocity description spaces, which can be expressed as Formula (10):(10)$$v_{r}=v_{m}\cap v_{d}\cap v_{a}$$

In the final velocity space, the evaluation function of the velocity pair can be determined, and the optimal motion trajectory can be selected through the evaluation function, as shown in Formulas (11)–(13).
(11)G(v,w)=max(a∗heading(v,m)+β∗dist(vm)+γ∗vel(v,m))
(12)$$heading\left(v,m\right)=1-\frac{\theta }{\pi }$$
(13)$$dist\left(v,m\right)==\left\{\begin{array}{c} \frac{d}{D},0\leq d\leq D\\ 1,d>D \end{array}\right.$$
where α,β,γ are three exponential scalar weighting coefficients, all between 0 and 1. θ represents the angle between the robot motion direction and the target point. The smaller the angle, the greater the value of the evaluation function, and the path is suitable for the motion direction. *D* represents the minimum distance between the simulated track and the obstacle, and *D* represents the preset maximum distance value. The farther the simulated trajectory is from the obstacle, the greater the value of this index and the greater the value of the evaluation function [15]; then, the path is suitable for the current motion direction of the robot. Finally, the velocity vector size of the simulated path is calculated as shown in Formula (14):(14)$$vel\left(v,m\right)=\frac{v}{v_{max}}$$
where *v* represents the linear velocity of the current motion track, and $v_{max}$ represents the maximum linear velocity in the dynamic window. The higher the motion speed, the greater the value of the index and the value of the evaluation function, indicating that the path is suitable for the current motion direction of the robot.

## 5. Experiment and Analysis

To evaluate the navigation system better and more comprehensively, we verified it on a simulation platform and in a real experiment. The simulation tool uses the gazebo simulation platform [16] of ROS to build different types of warehouse simulation. Figure 5, Figure 6 and Figure 7 show the indoor environment of the office, and the outdoor environments of the gas station and the narrow road in that order. The map built in the simulated narrow environment is shown in Figure 8, the actual navigation map is shown in Figure 9, and the path planning map is shown in Figure 10. The red line is the global planning route, and the blue line is the global planning route. The simulation used a turtlebot3 robot, and the laser lidar was a multiline lidar. The computer CPU had an Intel (R) core (TM), i5-10210u CPU @ 1.60 GHZ, Ubuntu version 18.04, and ROS version melodic.

To highlight the performance of the improved A* algorithm, simulation path planning experiments were conducted in offices, gas stations, and narrow channel scenes. The experimental results are shown in Table 1. Under the same target point location, the path planning time of the improved A* algorithm was shorter, indicating the superiority of the improvement.

The navigation software for the real environment was still based on the ROS system of Ubuntu 18.04. The 16 line laser radar was velodyne16, the depth camera was Intel realsense d435i, and the on-board computer was mic-770h-00a1. The motion model was the crawler difference model, and the built robot is shown in Figure 11. We compare the mapping speed of downsampled point clouds with that of upsampled point clouds. Three scenarios with great differences were selected for mapping. The offline mapping speed is shown in Table 2. It can be seen from the table that the downsampling filter processing can improve the offline mapping speed.

In the actual complex scene, compared with the two-dimensional laser radar, the robot designed by us was equipped with a multiline laser radar, which can well identify and avoid obstacles on the ground and met the detection requirements. The obstacles in the red area are shown in Figure 12.

We conducted a two-point repeatability accuracy test. In the real scene, we set the linear speed of the robot to 1 m/s and the angular speed to 1.5 rad/s. Set the position coordinates of the initial robot as (0, 0, 0), and the coordinates of the endpoint are shown in Table 3. A total of 50 groups were selected. The experimental results are shown in the table. The average error of X and Y was less than 5 cm, and the angle was less than 0.1 radian out—high accuracy. The error mainly came from the friction between the measurement and the ground. In order to test the navigation accuracy performance in the actual application state, we also conducted a multi-point repeated navigation accuracy test, selected four paths, and selected six target points on each path. In the multi-point repeated navigation accuracy test, the average error between the target point and the actual arrival position was less than 10 cm, and the angle error was less than 0.12 rad, which can meet the navigation requirements of the logistics robot in a warehouse environment.

## 6. Summary and Outlook

The use of intelligent inspection robots can improve the efficiency of logistics operations, provide an efficient workflow for the practical operation of traditional logistics warehousing, and greatly improve work efficiency. In this paper, a high-precision navigation system integrating multi-sensor information was designed for the complex scenes of real storage. To better realize the positioning in the disordered and ordered scenes, a 16-line laser lidar was used and the point cloud processing algorithm was designed, and the improved navigation algorithm was integrated for path planning. To better evaluate the navigation performance, simulation and real scene experiments were carried out at the same time. The results show that the navigation system can complete the navigation and positioning of complex scenes. Therefore, it is a new model for robot navigation and has important application significance. However, during the experiment, it was found that with the acceleration of speed, the performance of feature matching is greatly reduced. How to achieve navigation and positioning at a high speed will be the focuses and challenges of future research. 

## Figures and Tables

**Figure 1 sensors-22-07794-f001:**
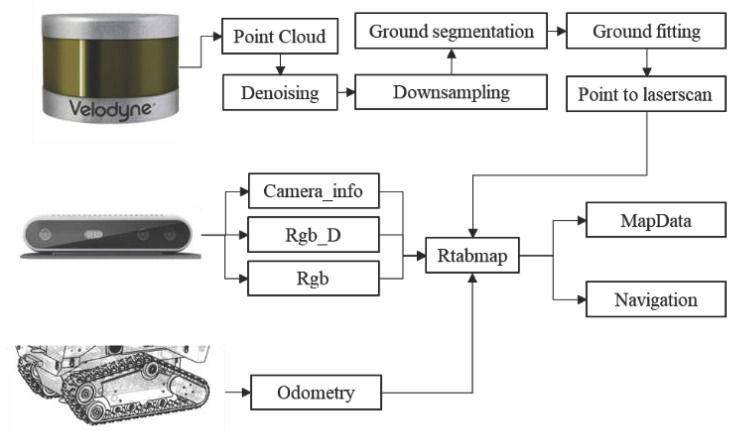
System framework.

**Figure 2 sensors-22-07794-f002:**
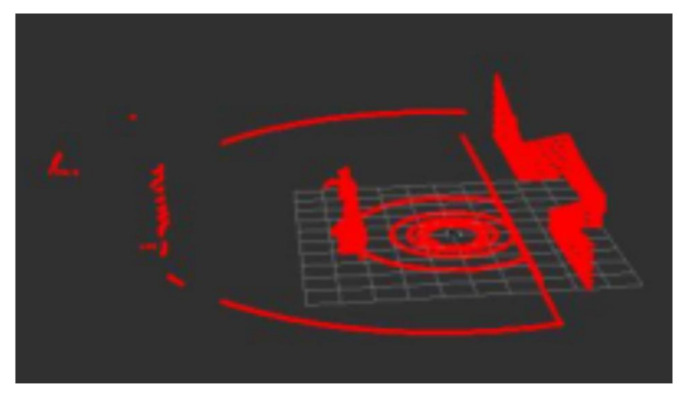
Original point cloud.

**Figure 3 sensors-22-07794-f003:**
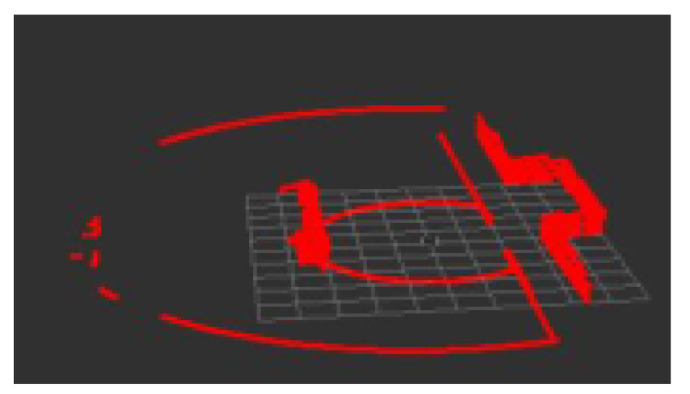
After denoising and desampling.

**Figure 4 sensors-22-07794-f004:**
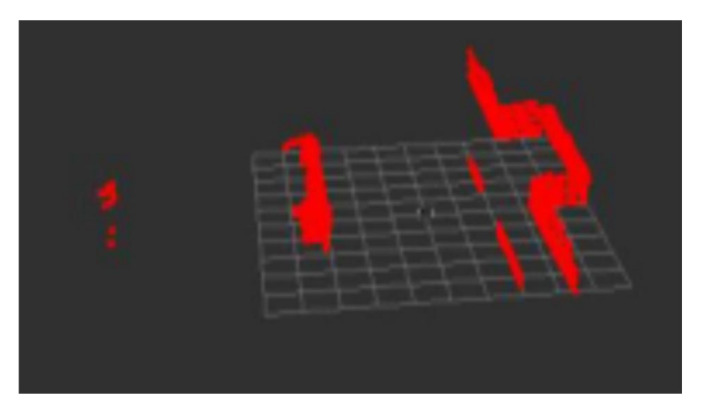
Point cloud map after ground segmentation, fitting, and conversion.

**Figure 5 sensors-22-07794-f005:**
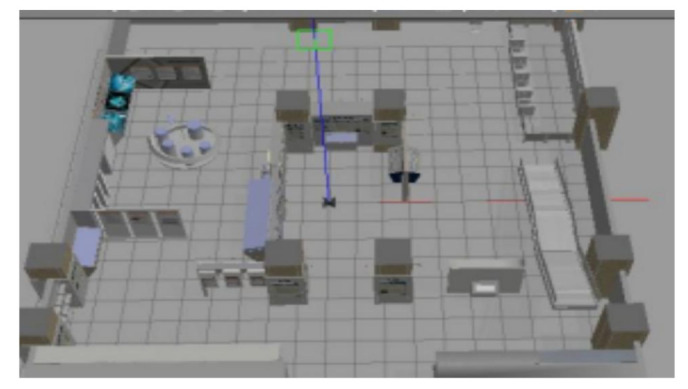
Office.

**Figure 6 sensors-22-07794-f006:**
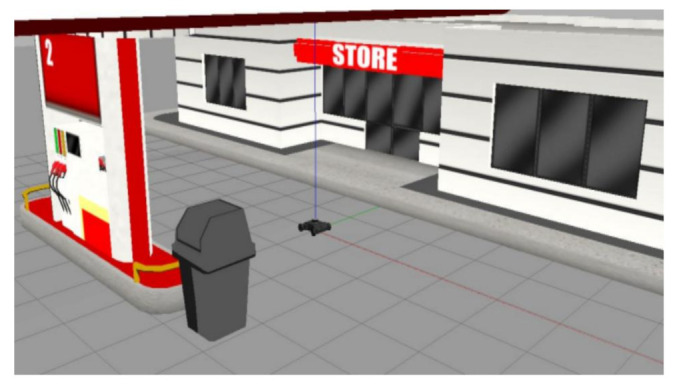
Gas station.

**Figure 7 sensors-22-07794-f007:**
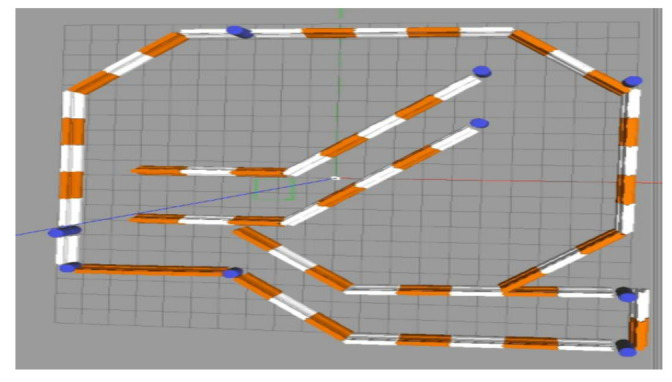
Narrow road.

**Figure 8 sensors-22-07794-f008:**
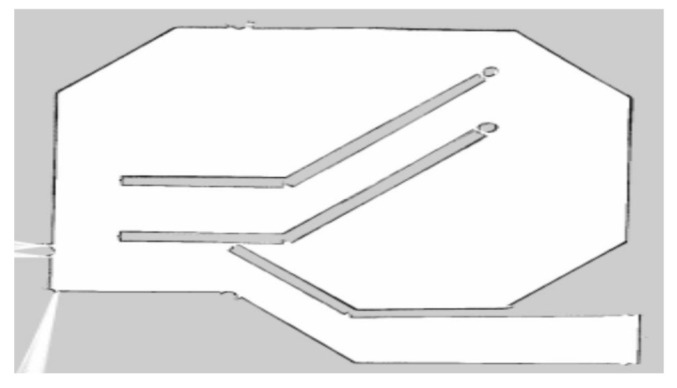
Scene map.

**Figure 9 sensors-22-07794-f009:**
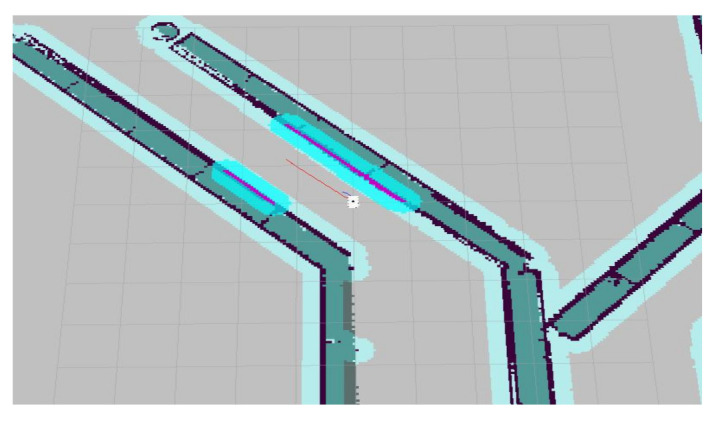
Navigation map.

**Figure 10 sensors-22-07794-f010:**
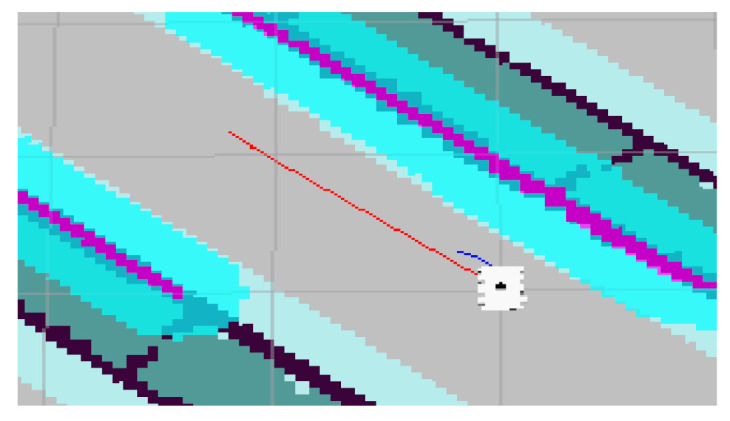
Navigation enlarged map.

**Figure 11 sensors-22-07794-f011:**
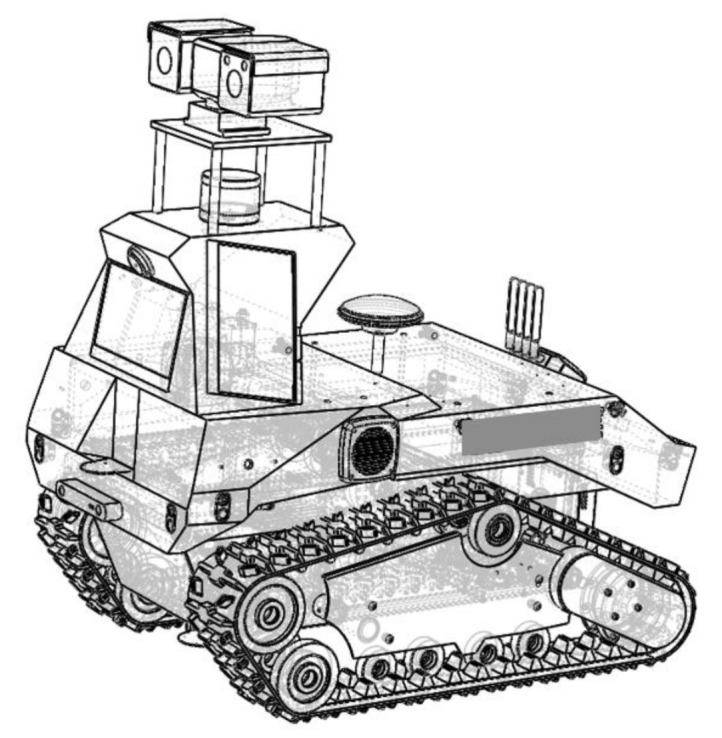
Logistics inspection robot.

**Figure 12 sensors-22-07794-f012:**
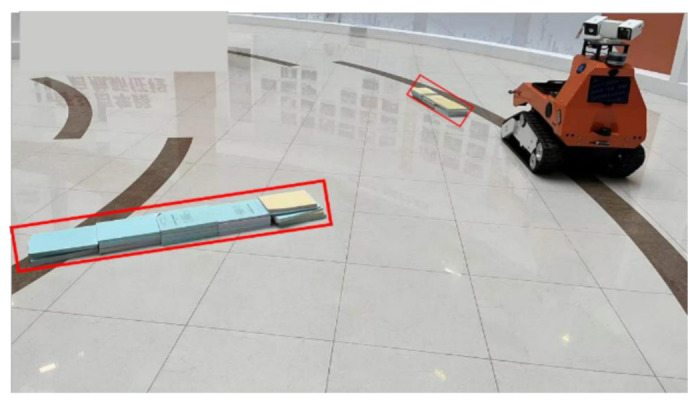
Obstacle scene.

**Table 1 sensors-22-07794-t001:** Planning time.

Scene	After Improvement(s)	Standard A* Algorithm(s)
Gas station	3	5
Office	6	9.2
Narrow road	2	3.9

**Table 2 sensors-22-07794-t002:** Comparison of mapping speed.

	Sense 1	Sense 2	Sense 3
Unsampling filter	307 s	210 s	502 s
Downsampling filter	270 s	192 s	454 s

**Table 3 sensors-22-07794-t003:** Partial navigation coordinates.

Initial	Set End	Actual Arrival	X	Y	Angular
Coordinates(m)	Coordinates(m)	Coordinates(m)	Error(m)	Error(m)	Error(rad)
0,0,0	3,3,3.14	2.99,3,3.13	0.01	0	0.01
0,0,0	6,6,3.14	5.98,5.99,3.14	0.02	0.01	0
0,0,0	9,9,3.14	9,8.98,3.12	0	0.02	0.02
3,3,0	0,0,3.14	0.01,0.01,3.13	0.01	0.01	0.01

## Data Availability

Not applicable.

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
