# Peer review of "The Navigation System of a Logistics Inspection Robot Based on Multi-Sensor Fusion in a Complex Storage Environment"

_sensors, 2022, doi:10.3390/s22207794_

Round 1

Reviewer 1 Report

This manuscript reports sensors fusion in the complex environment. The manuscript can be accpeted after the following issues are solved. 1, How to obtain the Figure 1? Are them from products or made by the authors?

2, Any reason for Figure 2-4? What is the mechanism for them?

3, As to the gas and block? any method for them in navigation?

4, Any data for the gas in data calculation?

5, To solid the background about the images, the author can cite more reference to strength the manuscrit, such as Micromachines, 2022, 13, 332.

Author Response

The response to the reviewer's comments is in the annex (PDF)

Reviewer 2 Report

The paper fails in providing any novel contribution.
The claim in the abstract "[...] this paper proposes a new robot navigation system based on vision and 3D radar information fusion [...]" is ambiguous, and it is contradicted by the fact that well-known techniques are employed. For the localization system, RTABMap, a well-known software, largely employed in the ROS community, is used. There is no novel contribution in pre-filtering / downsampling sensory data and passing it to RTABMap.
Regarding the navigation system, DWA is also a very well-known approach and it exists in several variants. The formulas described in the paper contain typos (e.g. (13)) and incorrect references (e.g. [10]). The system seems to use an old software implementation (ros melodic) of the DWA algorithm and no references are provided for more advanced versions of that algorithm, such as DWB in ROS2 (see: https://github.com/ros-planning/navigation2/blob/main/nav2_dwb_controller/README.md)

There are errors in both the figures and the tables numbers.

The word 'radar' means: `radio detection and ranging'. On the described robot no radio equipment is used, and this is misleading and confusing. Instead, the employed device is a 'lidar'. This error is repeated both in the abstract and in the main section.

In section 5, lines 122-124 contain some text (a leftover??) from the Microsoft Word template! Authors are recommended to check their manuscript before submitting it to a journal.
[...] Authors should discuss the results and how they can be interpreted from the perspective of previous studies and of the working hypotheses. The findings and their implications should be discussed in the broadest context possible. Future research directions may also be highlighted. [...]

Author Response

(The authors gave the same response as above.)

Reviewer 3 Report

According to the authors, the purpose of this paper was to design a mobile robot autonomous mapping and path planning system based on multi-sensor fusion, which obtains the surrounding environment information through the 3D lidar, and the autonomous positioning and mapping of the robot by using the real-time appearance mapping algorithm

The format of the paper is within the journal template, (Numbering, Citations system, Figure captions etc). The paper flows and has connectivity and clarity.

I can say that the article is well-organized, and the methodology developed is clearly explained.

According to the authors, they have differentiated themselves from existing studies-designs, and technologies by improving the autonomous navigation performance of mobile robots in complex scenes. If they carry out a small research on Google Scholar or various journals and specifically in https://www.mdpi.com/journal/sensors  there are similar studies recently published ( I could provide them if they want), providing more or less  the same context, consequently I would kindly suggest to the authors to emphasize and add some more information on where this study is different and unique

Author Response

(The authors gave the same response as above.)

Round 2

Reviewer 1 Report

This manuscript reported sensor fusion for robot navigation. The topic is interesting and useful. However, there are something needs further improve before publications. 

1, The introduction section is not strong, the authors can cite more reference paper to support the data confusion, such as micromachine, 2022, 13(2), 300.

2,Why the data de-nosing in figure 3? what the reeason to make pont cloud filtering?

3, can the authors draw the figures after the mapping and navigation? It seems that the author point out the algorithm but  without any figures?

4, as to the Figure 13, any novelty for your study? It seems that the noverlty of this manuscrip is not strong. The author can explain more about this part in the manuscript

Reviewer 2 Report

Review based on revision v3. Please check the English syntax and the punctuation accurately, the manuscript still contains many errors. 

Line 45 desampled -> downsampled

Line 53 "This paper selects" -> please check this sentence, the syntax is not clear

Line 69: "use the visual odometer to obtain the visual odometer" -> please check this sentence.

Line 74:
"In order to solve the problem that the traditional global path planning algorithm takes too long under the large-scale map with real storage."
First of all, the English syntax/punctuation is not correct (or something is missing).
Then, the authors should address the following points:
- what is the name of the "traditional global path planning algorithm".
- what kind of performance comparison has been done between the traditional algorithm and the modified A* solution proposed by the authors. To accept this statement, experimental results should be presented (e.g. given a map of 10 meters x 10 meters, with a cell resolution of 0.05 cm, algorithm A found a path in xxx seconds, compared to yyy seconds of algorithm B) 

 Section 3.2 does not explain how the global trajectory computed by the A* algorithm is processed by the local DWA navigator, i.e. how the two algorithms are interoperate.

Equation 10, on the right: it should be vm not vr

Line 96+: please explain exactly what is meant for "mapping speed". Is it the time required by RTABMAP for creating a map? Is it considering the time for the robot to move around in the environment? is it computed offline or online? etc.

Line 113: English check required

General consideration:
Please add the following reference. Additionally, the novelty of the proposed approach still needs to be carefully highlighted.
E. Marder-Eppstein, E. Berger, T. Foote, B. Gerkey and K. Konolige, "The Office Marathon: Robust navigation in an indoor office environment," 2010 IEEE International Conference on Robotics and Automation, 2010, pp. 300-307, doi: 10.1109/ROBOT.2010.5509725.

Author Response

Dear reviewer, the reply is in the appendix.

Round 3

Reviewer 1 Report

The authors did response all my issue. I recomment to publish in the current form

Author Response

Dear reviewer:
     Thank you for your decision.

Reviewer 2 Report

I recommend to have the paper checked again by a native speaker.
There are still major English issues.
L52-53
L74
L96
L108-111

I cannot find in the revised paper v3 any editing which refers to the data you are mentioning in your response:
"Inorder to highlight the performance of the improved A* algorithm, according to your suggestions, experimental verification has been added. Under the same experimental scenario, the improvedalgorithm has better performance. With the same target point, the path planning time of theimproved A * algorithm is 0.26s, while that of the standard A * algorithm is 0.5s.".
Data should be presented in a table, and also include a figure with the map and the full path (from the start to the goal) found by the A* algorithm. The map should also contain information about its dimensions and the cell size (in order to evaluate its complexity)

Table 2: The table indicates a sub-millimeter accuracy as the robot's final error in reaching the goal, which is absolutely not realistic.
The units are probably meters and not centimeters.

The authors should also indicate either:
- the results have been obtained using the gazebo simulator, under the hypothesis of perfect localization.
or:
- the results have been obtained with a real robot, but they do not take into account the localization error. Indeed, even if the algorithm reaches the goal, the real robot can be still several centimeters away from the expected position due to inaccuracies in the estimation of its position w.r.t. the map reference frame. This error cannot be easily evaluated, unless an external localization system is used (e.g. an external camera)

Author Response

We have completed the reply, see the attachment for details.
